# The association between childhood maltreatment and empathic perspective taking is moderated by the 5-HTT linked polymorphic region: Another example of "differential susceptibility"

**Vera Flasbeck**[1], **Dirk Moser**[2], **Johanna Pakusch**[3], **Robert Kumsta**[2], **Martin Brüne**[1]*

**1** LWL University Hospital, Department of Psychiatry, Psychotherapy and Preventive Medicine, Division of Social Neuropsychiatry and Evolutionary Medicine, Ruhr-University Bochum, Bochum, Germany,
**2** Department of Genetic Psychology, Faculty of Psychology, Ruhr-University Bochum, Bochum, Germany,
**3** Department of General Zoology and Neurobiology, Ruhr-University Bochum, Bochum, Germany

\* martin.bruene@rub.de

## Abstract

Previous research has suggested that the short (S)-allele of the 5-HT transporter gene-linked polymorphic region (5-HTTLPR) may confer "differential susceptibility" to environmental impact with regard to the expression of personality traits, depressivity and impulsivity. However, little is known about the role of 5-HTTLPR concerning the association between childhood adversity and empathy. Here, we analyzed samples of 137 healthy participants and 142 individuals diagnosed with borderline personality disorder (BPD) focusing on the 5-HTTLPR genotype (S/L-carrier) and A/G SNP (rs25531), in relation to childhood maltreatment and empathy traits. Whereas no between-group difference in 5-HTTLPR genotype distribution emerged, the S-allele selectively moderated the impact of childhood maltreatment on empathic perspective taking, whereby low scores in childhood trauma were associated with superior perspective taking. In contrast, L-homozygotes seemed to be largely unresponsive to variation in environmental conditions in relation to empathy, suggesting that the S-allele confers "differential susceptibility". Moreover, a moderation analysis and tests for differential susceptibility yielded similar results when transcriptional activity of the serotonin transporter gene was taken into account. In conclusion, our findings suggest that the S-allele of the 5-HTTLPR is responsive to early developmental contingencies for "better and worse", i.e. conferring genetic plasticity, especially with regard to processes involving emotional resonance.

## Introduction

Social interaction requires reciprocal understanding of verbal and nonverbal signals, which entails the ability to understand others' emotional states. The mechanism involved in this

**Data Availability Statement:** All relevant data are within the paper and its Supporting Information files.

**Funding:** The author(s) received no specific funding for this work.

**Competing interests:** The authors have declared that no competing interests exist.

process is commonly referred to as "empathy", which can be conceptualized as a semi-automatic sharing of another's feelings, combined with the ability to differentiate between own and others' affect [1–3]. Empathic deficits have been described in several neuropsychiatric conditions including autism [4, 5], schizophrenia [6, 7], psychopathy [8] and personality disorders, with mixed results for borderline personality disorder [9–13]. Our own research revealed that patients with borderline personality disorder (BPD) showed selectively increased empathy for psychological pain compared to somatic pain [14], which was associated with childhood trauma, alexithymia and emotional empathy.

A plethora of studies suggests that the activity of the serotonergic system is critically involved in social behavior [15, 16]. With regard to gene-environment interaction, for instance, research in nonhuman primates demonstrated that length variations located in the promoter of the serotonin transporter gene (5-HTTLPR) affects, together with rearing experiences, the level of serotonin metabolites in the cerebrospinal fluid and primate social behavior [17, 18]. In humans, the short (S) allele of the 5-HTTLPR has been associated with reduced serotonin transporter expression and function; it has also been found to be related to trait-anxiety, depression and impulsivity [19–23]. With regard to the processing of social cues, previous studies reported increased emotional reactivity, especially towards negatively biased stimuli [24–28] as well as heightened physiological stress responses in carriers of the S-allele [27, 29]. However, this polymorphic variation does not generally occur more frequently in clinical samples compared to the general population [30–32]. The general population is heterozygous, whereas the LL-genotype is less common, and the SS-variant relatively rare, in part depending on ethnicity [33, 34]. In addition, controversy exists about the effect of rs25531, a SNP within the 5-HTTLPR repetitive element, with the A-variant of the L-allele being associated with greater transcriptional activity and thus more efficient serotonin turnover [35–37], whereby a linkage disequilibrium between 5-HTTLPR and rs25531 has been described, with the rarer G-variant of rs25531 occurring more frequently together with the L-allele than with the S-allele [36, 38]. In recent years, researchers have become interested in the question, raised from an evolutionary point of view, why genes conferring increased risk to psychological dysfunction may be conserved in the genepool of human populations or have even been positively selected in recent millennia (see, for instance, [39]). As an alternative account to the widely-known "diathesis-stress-model" [40], the groups of Ellis [41] as well as Belsky and colleagues have suggested that genetic variants may not one-sidedly convey risk to the development of psychological dysfunction if associated with adverse life events, but that the very same polymorphic variation may confer lower than average risks if met with superior environmental conditions, foremost empathic parental care and emotional availability of care-givers [42]. Therefore, the "differential susceptibility" or "genetic plasticity" model emphasizes the difference between plasticity and resilience (i.e. unresponsiveness to environmental conditions; [43]).

With regard to the 5-HTTLPR, several studies reported an association of stressful live events and depression in SS-homozygotes or S-carrying heterozygotes ([44]; for meta-analyses, see [45, 46]), while others did not confirm these findings (for meta-analyses, see [31, 47]). Conversely, and in line with the "differential susceptibility" model, Pluess and colleagues reported that SS-carriers had higher scores in neuroticism when exposed to negative life events (within the last six months), whereas more positive life events were related to less than average neuroticism. This association was absent in L-carriers [48]. Similar findings were reported by Kuepper et al. [49] who also found an association of negative life events (over the life span) and neuroticism in S-allele carriers.

However, whether or not traumatic events during childhood specifically influence the development of empathic abilities, moderated by the 5-HTTLPR, is still unclear. Accordingly, the present study aimed to investigate the impact of the 5-HTTLPR on trait empathy in a

sample of healthy participants and patients with BPD. We deliberately chose the two samples, because one was characterized by relatively few adverse childhood experiences, while the clinical group was coined by relatively high degrees of early maltreatment. We specifically hypothesized that the S-allele of the 5-HTTLPR would differentially impact on the association of childhood trauma with empathic perspective taking, whereas the L-allele would be unresponsive to environmental variation, with some potential modification according to the transcriptional activity of the serotonin transporter gene conveyed by the rs25531 polymorphism.

## Material and methods

### Participants

For the current study we recruited female in-patients with BPD, diagnosed according to DSM-5 criteria [50] from the LWL-University Hospital Bochum and female healthy control participants via advertisement. In total, 142 patients with BPD and 137 control participants were included. The age of participants was between 18 and 50 years. All participants were fluent in German, free of somatic illnesses and not pregnant (see Table 1 for comorbid disorders and medication of BPD patients). The control participants were free of medication and psychiatric disorders. Regarding the ethnical background, 95.2% were Caucasians, 4.4% originated from the Middle East (mainly of Turkish origin) and 0.4% from the Far East (Vietnamese). The study was approved by the Ethics Committee of the Medical Faculty of the Ruhr-University Bochum (project number 4639–13). The authors assert that all procedures contributing to this work comply with the ethical standards of the relevant national and institutional committees on human experimentation and with the Helsinki Declaration of 1975, as revised in 2008. All participants gave their full informed consent in writing.

### Questionnaires

Premorbid or general intelligence was estimated using the Mehrfachwahl-Wortschatz-Intelligenz-Test (MWT-A; [51]). Empathic abilities were measures using the German version of the Interpersonal Reactivity Index [52], called "Saarbrücker Persönlichkeits-Fragebogen" [53]. This questionnaire comprises four subscales, namely "perspective taking" (PT), "fantasy" (FS), "'empathic concern" (EC) and "personal distress" (PD), and has proven reliable with a Cronbach's alpha of 0.78. The present analysis focused on the "perspective-taking" score of the Interpersonal Reactivity Index (IRI), because this score is suggested to reflect cognitive

**Table 1. Comorbid disorders and medication of patients with BPD in absolute (n) and relative (in %) amounts.**

|  | n | % |
|---|---|---|
| **Comorbid disorders of patients with BPD** |  |  |
| Depressive episode | 74 | 52.1 |
| Posttraumatic Stress Disorder | 21 | 14.8 |
| Phobic/ anxiety Disorder | 7 | 4.9 |
| Substance misuse | 42 | 29.8 |
| **Medication** |  |  |
| without regular medication | 59 | 41.5 |
| antidepressant | 51 | 35.9 |
| antipsychotic | 22 | 15.5 |
| antidepressant and antipsychotic drugs | 22 | 15.5 |
| antiepileptic | 8 | 5.6 |
| Other psychoactive drugs | 6 | 4.2 |

empathy traits. In contrast, emotional empathy is more context dependent [54] and therefore emotional empathy scores are not appropriate for trait analyses (the validity of the other cognitive empathy score of the IRI, "fantasy", is debated and therefore not included into the present analyses; [55]).

The short German version of the Childhood Trauma Questionnaire (CTQ) was used to assess the experience of maltreatment during childhood. The CTQ contains 28 questions tapping into the history of emotional abuse, physical abuse, sexual abuse, emotional neglect and physical neglect. Participants were asked to rate the occurrence of maltreatment on a 5-point Likert scale (1 = never, 5 = very often) pertaining to their childhood and youth. Cronbach's alpha values for the German version were high for all subscales (0.80), except for physical neglect [56]. In addition, we used the Beck's Depression Inventory to assess the self-rated level of depressivity [57].

## Genotyping

The DNA samples of participants were collected using Oragene OG-500 collection kits (DNA Genotek, Inc., Ottawa, ON, Canada) and by mouthwash with a commercially available mouthwash solution (Listerine). The DNA extraction was conducted according to the manufacturer's instructions of the Oragene Kit and by an adapted version for the mouthwash samples, using a standard salting-out procedure proposed by Miller et al. [58]. The DNA samples were diluted to a concentration of (20 ng/μL). The 5-HTTLPR and the rs25531 genotypes were determined as described by Wendland et al. [36].

## Statistical analyses

We conducted a power analysis for interaction effects, i.e. the differences between slopes for the moderation model with the genotypes SS+SL and LL, using G*Power, Version 3.1.9.2. [59]. Power calculation for the current sample of 205 participants was determined by the following model: t-test-linear bivariate regression, two groups, difference between slopes, with α set at 0.05. Standard deviation of the residuals, the sample size and the difference between the slopes were also considered. Accordingly, the statistical power coefficient was 0.75.

In accordance with previous studies, we divided the sample into S-carriers (SS+SL pooled) and LL-carriers (e.g. [60–62]). This approach was justified, because previous studies reported no differences between SS and SL-carriers with regard to personality traits, suggesting a dominant-recessive type of association of the S-allele with personality (21).

Similarly, following previous research (e.g., [37]), subjects were further divided into groups according to the "transcriptional activity" (TA) of the rs25531.

Independent two-sample t-tests were used for comparisons of questionnaire data between groups. The distributions of genotypes were assessed by chi-square tests, whereas calculations were performed for SS, SL and LL genotypes, and for both groups, i.e. patients and controls. Since we did not find any effect of group for the genotype distribution, further analyses were performed without the factor group. Because of the difference between groups regarding age and IQ, we included these variables as covariates into further analyses. In order to investigate the effect of the genotype, we calculated a multivariate analysis of covariance (MANCOVA) with the covariates IQ and age and the between-subject factor genotype (SS+SL vs. LL) and the independent variables were the IRI scores. The moderation analysis was conducted by means of the SPSS macro tool PROCESS developed by Hayes [63]. The moderation was calculated for the dependent variable "perspective taking" (Y; outcome) and the independent variable CTQ total score (X; predictor) and the moderator (M; susceptibility factor), i.e. the 5-HTTLPR genotype (SS+SL vs. LL), as well as controlled for IQ and age (Table 2).

In order to explore whether the data were in accordance with the differential susceptibility model, we further investigated the association of the predictor (CTQ) and the susceptibility factor, the moderator, by correlation analyses (partial correlations corrected for age and IQ between CTQ total score and the genotype). In addition, we tested for an association of the susceptibility factor with the outcome variable by calculating the correlation of genotype with perspective taking (partial correlation). Finally, we calculated the difference between the slopes of the associations of the moderation analysis between the two genotype groups.

The same analyses for moderating effects and differential susceptibility were also carried out according to differences in transcriptional activity. Additional moderation analyses were performed for the SS, SL and LL Genotypes and for group (BPD vs. HC) as the moderator instead of Genotype. In order to examine the impact of depressivity, we calculated additional moderation analyses with the additional covariate "BDI score" for the moderators "SS+SL vs. LL", "SS, SL and LL" and transcriptional activity (Table 2). The statistical analyses were performed using the software SPSS version 25 (IBM Corp. Released 2017. IBM SPSS Statistics for Windows, Version 25.0. Armonk, NY: IBM Corp.).

## Results

### Questionnaires

Patients differed in age and IQ from control participants and reported more severe experiences of childhood maltreatment as well as more depressive symptoms. Patients reached higher scores in the personal distress score of the IRI, whereas healthy controls scored higher in perspective taking and fantasy scores (Table 3).

### Genotypes

The distributions of genotypes were in Hardy-Weinberg equilibrium in the whole sample ($X^2$ = 0.096; $p$ = 0.757) and were distributed as follows: LL genotype n = 103 (34.2%), SL genotype n = 131 (43.5%); SS genotype n = 45 (15%). When groups were analyzed separately, the distributions were also in Hardy-Weinberg equilibrium (BPD: $X^2$ = 0.23; $p$ = 0.631; HC: $X^2$ = 0.02; $p$ = 0.887) and there was no difference between allelic frequencies between groups ($X^2$ = 2.84; $p$ = 0.241; df = 2; Table 4). The division into S or LL-carriers resulted in n = 103 LL-carriers and n = 176 S-carriers. Another group formation was based on the rs25531, which offered the opportunity to form groups based on "transcriptional activity"(TA) (Table 4).

### Questionnaires and genotypes

The MANCOVA for the empathy scores revealed main effects for age and IQ (age $F(4, 246)$ = 3.48, $p$ = 0.009; IQ $F(4, 246)$ = 6.69, $p < 0.001$), but no main effect of genotype or interaction with genotype.

**Table 2. Summary of moderation analyses conducted.** The predictor and outcome variables remained constant across calculations whereas the moderator and control variables were exchanged.

| Predictor | Outcome | Moderator | Covariates |
|---|---|---|---|
| CTQ total score | Perspective taking | SS+SL vs. LL | age, IQ |
| | | | age, IQ, BDI |
| | | SS vs. SL vs. LL | age, IQ |
| | | | age, IQ, BDI |
| | | TA groups: low/low, high/low, high/high | age, IQ |
| | | | age, IQ, BDI |
| | | BPD vs. HC | age, IQ |

**Table 3. Psychometric properties of patients with BPD and healthy participants.** Results are reported as mean (M) values and standard deviations (SD). *t*, *p* and *df* of T tests between groups are shown.

| | HC | | BPD | | T Test | | |
|---|---|---|---|---|---|---|---|
| | *M* | *SD* | *M* | *SD* | *t* | *p* | *df* |
| Age | 24.8 | 5.6 | 27.6 | 7.9 | 3.41 | 0.001 | 253.6 |
| IQ | 108.8 | 17.1 | 101.4 | 16.9 | -3.48 | 0.001 | 254 |
| BDI | 5.9 | 5.9 | 35.3 | 10.3 | 28.33 | <0.001 | 207.4 |
| Childhood Trauma Questionnaire | | | | | | | |
| Total score | 33.6 | 10.7 | 63.7 | 19.5 | 14.11 | <0.001 | 166.3 |
| Emotional abuse | 7.5 | 3.7 | 17.1 | 5.8 | 14.50 | <0.001 | 182.9 |
| Physical abuse | 5.6 | 2.1 | 9.6 | 5.2 | 7.41 | <0.001 | 139.4 |
| Sexual abuse | 5.3 | 1.1 | 9.5 | 5.9 | 7.40 | <0.001 | 115.1 |
| Emotional neglect | 8.5 | 3.9 | 17.2 | 5.7 | 13.21 | <0.001 | 190.7 |
| Physical neglect | 6.2 | 2.1 | 10.3 | 4.3 | 8.85 | <0.001 | 158.2 |
| Interpersonal Reactivity Index | | | | | | | |
| Perspective taking | 19.3 | 4.3 | 13.8 | 5.97 | -8.75 | <0.001 | 221.7 |
| Fantasy | 19.1 | 5.5 | 16.4 | 7.0 | -3.44 | 0.001 | 224.7 |
| Empathic concern | 20.4 | 4.1 | 19.8 | 5.4 | -0.90 | 0.375 | 221.1 |
| Personal distress | 12.8 | 5.1 | 21.6 | 4.2 | 15.03 | <0.001 | 249.4 |

## Moderation analyses

In order to examine the nature of formation of specific social behavior associated with the 5-HTTLPR-genotype, we performed moderation analyses: We calculated the moderation for the independent variable CTQ total score (X; predictor), the dependent variable perspective taking (Y; outcome) and the moderator (M; susceptibility factor) the 5-HTTLPR genotype (SS+SL and LL). The overall model was significant ($F(5, 199) = 6.48$, $p < 0.001$, $R^2 = 0.1401$), as was the interaction of CTQ by genotype (Interaction $b = -0.0810$, $t(199) = -2.16$, $p = 0.032$). According to Belsky et al. [64] and our own previous work [65], we investigated whether the data were compatible with the idea of differential susceptibility. Accordingly, we first tested if the predictor was associated with the moderator. Here, no correlation emerged between CTQ score and genotype ($r = -0.130$, $p = 0.062$). Second, the susceptibility factor (genotype) did not

**Table 4. Overview of 5-HTTLPR genotype distributions in the whole sample, and for patients with BPD and healthy controls (HC) separately.** The first column shows the distribution of the 5-HTTLPR genotypes regarding SS and LL homozygotes and SL heterozygotes. The second column shows the group formation according to the 5-HTTLPR and rs25531 genotypes, which results in high/high, high/low and low/low transcriptional activity groups.

| 5-HTTLPR genotypes | | | | Transcriptional activity groups | | | | |
|---|---|---|---|---|---|---|---|---|
| | n total | BPD | HC | | | n total | BPD | HC |
| LL | 103 | 59 | 44 | high/ high | | 75 | 44 | 31 |
| | | | | | $L_AL_A$ | 75 | 44 | 31 |
| SL | 131 | 63 | 68 | | | | | |
| | | | | high/low | | 140 | 68 | 72 |
| SS | 45 | 20 | 25 | | $L_AL_G$ | 27 | 15 | 12 |
| | | | | | $L_AS_A$ | 112 | 52 | 60 |
| | | | | | $L_AS_G$ | 1 | 1 | 0 |
| | | | | low/ low | | 63 | 28 | 33 |
| | | | | | $L_GL_G$ | 1 | 0 | 1 |
| | | | | | $L_GS_A$ | 17 | 10 | 7 |
| | | | | | $S_AS_A$ | 45 | 20 | 25 |

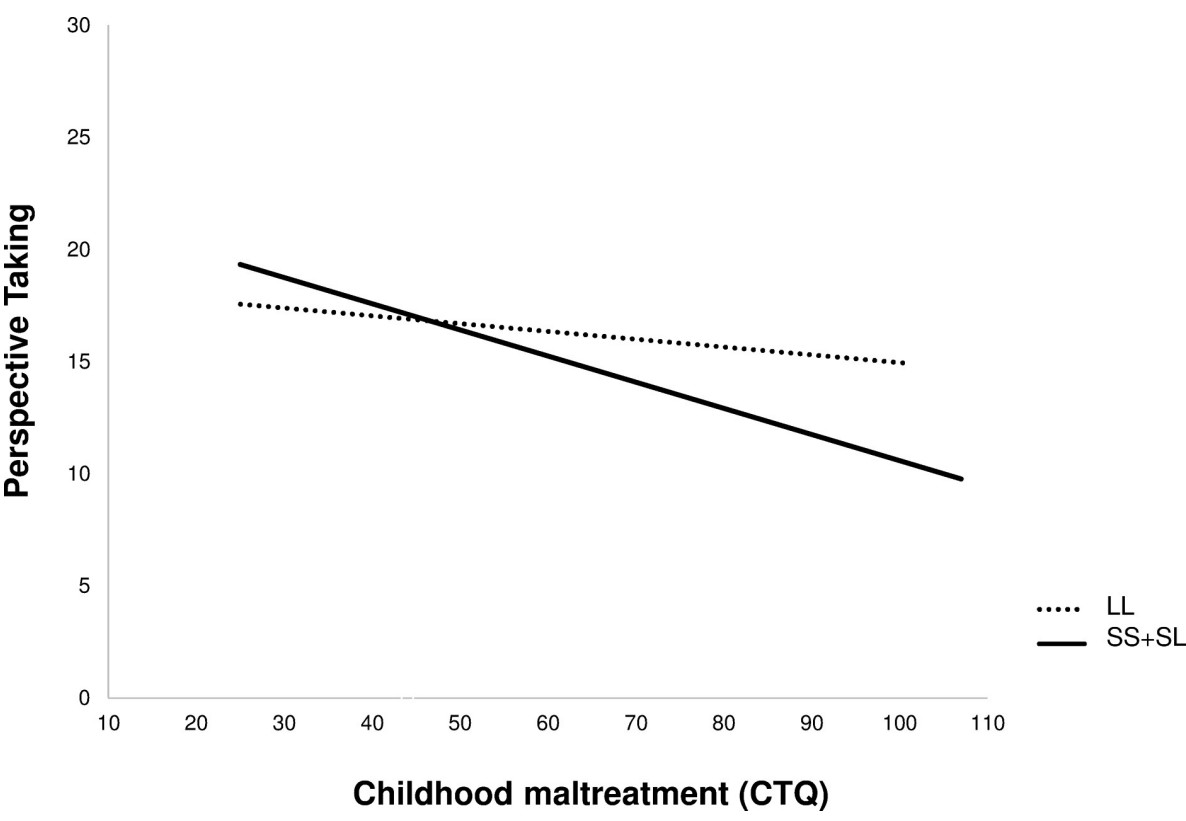

**Fig 1.** Comparison of the regression lines of S-carriers (solid line) and LL-carriers (dashed line). The diagram supports the notion of differential susceptibly showing the crossing of the lines with the simple slopes differing between genotypes.

correlate with the outcome parameter (perspective taking; $r = 0.084$, $p = 0.183$). Third, we checked whether the simple slopes of the associations of CTQ with perspective taking differed significantly from zero. The significant difference to zero was only present for the slope of the regression in S-carriers (SS+SL $b = -0.109$, SE = 0.023, $p < 0.001$, LL: $b = -0.028$, SE = 0.031, $p = 0.378$; Fig 1; Table 5). Forth, we compared the simple slopes and found a significant difference between the groups with $t = 3.08$, $p = 0.002$ (SS+SL vs LL). In sum, these analyses are compatible with the "differential susceptibly" model, suggesting that the S-genotype may confer genetic plasticity to environmental variation.

We also calculated a moderation analysis for the three genotypes, SS, SL and LL. Here, the model was highly significant ($F(5, 199) = 6.77$, $p < 0.001$, $R^2 = 0.1374$). However, the interaction of CTQ by genotype showed only a tendency toward statistical significance (Interaction $b = -0.0497$, $t(199) = -1.84$, $p = 0.067$), which could be related to the relatively small sample in the SS group (for comparison see Table 4).

In order to investigate the impact of the rs25531 within the 5-HTTLPR we built "transcriptional activity" (TA) groups (Table 4). We re-calculated the moderation analysis for the independent variable CTQ total score (X; predictor), the dependent variable perspective taking (Y; outcome) and the TA group as the moderator (M; low/low, high/low and high/high) The overall model was significant with ($F(5, 199) = 6.77$, $p < 0.001$, $R^2 = 0.1453$), as was the interaction of CTQ by genotype (Interaction $b = -0.0624$, $t(199) = -2.44$, $p = 0.015$). Moreover, no correlations were found between CTQ score and TA group ($r = -0.108$, $p = 0.123$) or between perspective taking and TA group ($r = 0.094$, $p = 0.138$). Next, the analysis of slopes showed that only the slopes of TA groups high/low and low/low differed significantly from 0 (high/high: $b =$

**Table 5. Summary of moderation analyses performed for the predictor variable "CTQ" (total score) and the outcome variable "perspective taking".** The table shows the moderator variables and covariates used and the respective model statistics and interactions between the predictor and moderator variables. The last two columns show tests for differential susceptibility, i.e. differences between slopes and differences from zero.

| Moderator | Covariates | Model | Interaction moderator*CTQ | Differences of slopes from zero | Differences between slopes |
|---|---|---|---|---|---|
| SS+SL vs. LL | age, IQ | $F(5, 199) = 6.48$, $p < 0.001$, $R^2 = 0.1401$ | $b = -0.0810$, $t(199) = -2.16$, $p = 0.032$ | SS+SL $b = -0.109$, SE = 0.023, $p < 0.001$ | $t = 3.08$, $p = 0.002$ |
| | | | | LL: $b = -0.028$, SE = 0.031, p = 0.378 | |
| | age, IQ, BDI | $(F(6, 196) = 9.20$, $p < 0.001$, $R^2 = 0.2198$ | $b = -0.0857$, $t(196) = -2.38$, $p = 0.018$ | | |
| SS vs. SL vs. LL | age, IQ | $F(5, 199) = 6.77$, $p < 0.001$, $R^2 = 0.1374$ | $b = -0.0497$, $t(199) = -1.84$, $p = 0.067$ | | |
| | age, IQ, BDI | | $b = -0.0479$, $t(196) = -1.84$, $p = 0.067$ | | |
| TA groups: low/low, high/low, high/high | age, IQ | $F(5, 199) = 6.77$, $p < 0.001$, $R^2 = 0.1453$ | $b = -0.0624$, $t(199) = -2.44$, $p = 0.015$ | high/high: $b = -0.0378$, SE = 0.0254, $p = 0.1389$ | high/high vs. low/low: $t = 3.30$, $p = 0.001$ |
| | | | | high/low: $b = -0.0808$, SE = 0.0189, $p < 0.001$; | high/low vs. low/low: t = 2.00, p = 0.047 |
| | | | | low/low: $b = -0.1237$, SE = 0.0262, $p < 0.001$ | high/high vs. high/low group t = 2.01, p = 0.046 |
| | age, IQ, BDI | $F(6, 196) = 9.43$, $p < 0.001$, $R^2 = 0.2239$ | $b = -0.0649$, $t(196) = -2.63$, $p = 0.009$ | | |
| BPD vs. HC | age, IQ | $F(5, 199) = 11.71$, $p < 0.001$, $R^2 = 0.2272$ | $b = 0.0437$, $t(199) = 0.81$, $p = 0.419$ | | |

-0.0378, SE = 0.0254, $p = 0.1389$, high/low: $b = -0.0808$, SE = 0.0189, $p < 0.001$; low/low: $b = -0.1237$, SE = 0.0262, $p < 0.001$). The comparisons of slopes revealed that the slopes of the regression lines of high/high and high/low groups were significantly different from the slope of the low/low group (high/high vs. low/low: $t = 3.30$, $p = 0.001$; high/low vs. low/low: $t = 2.00$, $p = 0.047$). The slope of high/high also differed from the slope of high/low group ($t = 2.01$, $p = 0.046$; see Fig 2). These results confirm the differential susceptibility model and extend our results to the level of transcriptional activity.

We further aimed to examine whether the association of childhood trauma with perspective talking was related to diagnosis. Therefore, we performed the same moderation analysis, but tested the factor "group" (BPD vs. HC) as the moderator. The model was also significant ($F(5, 199) = 11.71$, $p < 0.001$, $R^2 = 0.2272$), but the interaction of CTQ by group was not (Interaction $b = 0.0437$, $t(199) = 0.81$, $p = 0.419$), suggesting that a diagnosis of BPD was not the sole factor impacting on the association of childhood trauma with empathic perspective taking. Since depressivity is highly prevalent among individuals suffering from BPD, we repeated the moderation analyses and included also the BDI score as a covariate into the calculation. Here, the same results were obtained as in the previous analyses without the covariate BDI (see Table 5).

## Discussion

In the present study we aimed to investigate the role of the 5-HTLLPR concerning the association between childhood trauma and trait empathy. The moderation analysis and tests for differential susceptibility showed that the influence of childhood maltreatment on empathic perspective-taking seemed to be specific for S-carriers, suggesting that this allele confers genetic plasticity to environmental variation. Put differently, in people with at least one S-allele childhood maltreatment seems to be related to reduced perspective taking, whereas the

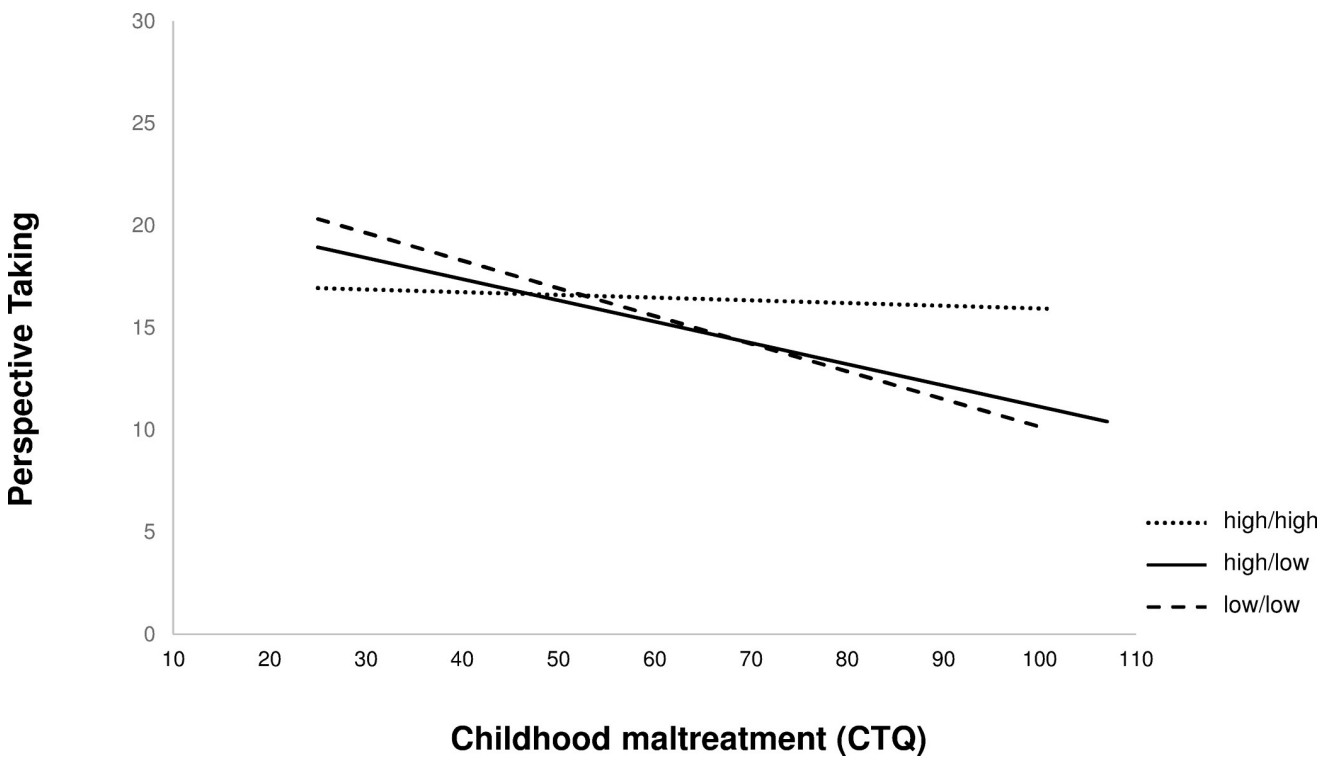

**Fig 2.** Graphical representation of the association of childhood maltreatment (CTQ) and perspective taking ability in high/high (small dashed line), high/low (solid line) and low/low (longer dashed line) transcriptional activity groups based on 5-HTTLPR and rs25531.

absence of childhood maltreatment is associated with well-preserved perspective taking capacities. As noted by Belsky et al. [66], the mere absence of maltreatment is not equivalent to high-quality parenting. When applying this idea to the present findings, it can tentatively be hypothesized that emotional warmth and availability during early development may lead to even better-than-average perspective-taking abilities (in this case, the slopes of the regression lines shown in Fig 1 may diverge to a greater extent if extrapolated to the left). In contrast, LL-carriers seemed to be unresponsive to childhood adversity in terms of consequences for trait empathy. The relation of trauma and reduced perspective-taking was already shown in previous studies and shown to be related to alexithymia and stress [14]. A possible explanation for reduced perspective taking in S-allele carriers exposed to childhood trauma may be the negative effect of stress, induced by the traumatic history and the following consequences (e.g. unsuccessful coping strategies), on the development of cognitive empathic perspective taking. Our findings are consistent with previous studies reporting that S-carriers showed increased attention towards emotional stimuli and especially negative stimuli when compared to L-carriers [24, 28, 67]. Owens and colleagues further reported that S-homozygotes (adolescents) were impaired in emotion recognition of negative and neutral stimuli and had more difficulties in responding to ambiguous negative feedback [67].

When looking at prosocial behavior, Stoltenberg and colleagues reported that S-allele carriers scored higher in social anxiety and lower in prosocial behavior [62]. Moreover, lower levels of sensitive responsiveness to their own toddlers were found in parents with the SS-allele [42]. Unfortunately, these studies did not include measures of the participants' own experiences during childhood. Moreover, Gyurak et al. [27] found that SS-homozygotes showed greater levels of emotional reactivity accompanied by an increased psychosocial stress response.

Similarly, another group reported that SS-carrier showed the greatest increase in cortisol levels following the Trier Social Stress Test. In addition, the association between genotype and cortisol reactivity was strongest when receiving negative feedback. The authors concluded that carrying the SS-allele may make the individuals more vulnerable to stressful life events, which leads to a greater risk for the severe psychological and physical health consequences associated with heightened cortisol exposure [29]. In support of this assumption Gotlib and colleagues examined the association between stress, 5-HTTLPR and depression in children. They demonstrated that girls, who were homozygous for the S-allele showed higher and prolonged cortisol levels in response to a stressor (mental arithmetic and Ewart Social Competence Interview) compared to L-allele carrying girls [68]. Additionally, it was shown that acute stress exposure led to a significant impairment in the inhibition of negative affective information only in SS-carriers. The authors concluded that a cognitive-attentional bias for negative emotional information may make an individual more vulnerable for stress-induced depressive symptoms [69]. With regard to our study, increased stress-reactivity and altered emotion processing in S-allele carriers may cause, together with the experiences of childhood adversity, impairment in taking the perspective of another individual. With regard to depression, the exact role of the 5-HTTLPR in the development of depression is unclear. This was shown by recent meta-analyses, which reported an association of stressful life events and depression in SS-homozygotes or S-carrying heterozygotes [45, 46], whereas other studies failed to determine such an association [31, 47]. For example, Culverhouse and colleagues found a significant main effect of sex and life stressor (high risk factor for depression), but they did not found an effect of genotype on the association of stress and depression, even if they included only studies with large sample sizes [47]. The authors concluded that, if any interaction would exist, it would not be a generalizable effect, only detectable in limited situations and of modest effect size. In our study, the inclusion of depressivity as a covariate did not affect the interaction, which suggests that the effect on perspective taking was not due to depressive symptoms. This further implies that our study does also not support the link between 5-HTTLPR, stress and depression. Eventually, the 5-HTTLPR, together with stress, may induce stress and emotion processing impairments (as described above), which lead only in a subset of individuals to the development of depression. This subset may bear additional risks, which are currently not in the focus of interaction studies. One potential factor could be the transcriptional activity of the serotonin transporter gene, which could be altered by epigenetic processes or other SNPs, as for example the rs25531 [35, 36, 70].

In our study, additional analyses according to the transcriptional activity of the serotonin transporter gene revealed similar results, i.e. differential susceptibility in the low/low and high/low groups akin to what emerged in S-allele carriers of the 5-HTTLPR. This finding is in accordance with previous studies, which also did not report modulation of the associations between 5-HTTLPR and phenotypes by the rs25531 [37, 71]. Interestingly, however, we found a graded effect of the transcriptional activity, indicating that the lower the activity of the serotonin transporter gene, the greater the genetic plasticity with regard to the effect of childhood adversity on empathic perspective-taking. When calculating the additional moderation analysis with the three 5-HTTLPR groups, SS, SL and LL, the interaction failed to reach significance. This could be due to a lack of statistical power due to the small sample size in the SS-group, or be related to the fact that the exact genotype may not be as relevant as the existence of at least one "risk allele". In any event, even though the statistical power for detecting significant interaction effects was sufficiently large with $\alpha = 0.75$, it is warranted to replicate the study in an independent, and preferably larger, sample.

Our study has several other limitations. For one, since we included only female participants, our conclusions cannot be generalized for males. Second, the clinical and the non-clinical

group differed significantly with regard to the experience of childhood trauma. However, since our main interest pertained to the influence of childhood adversity on trait empathy and its moderation by genotype, we were much less concerned with the presence or absence of specific effects of a diagnosis of BPD. In support of this idea, the moderation analysis with the moderator group (BPD/HC) did not show a specific effect of BPD on the association of childhood trauma with perspective taking. Third, as already pointed out, for a more substantial corroboration of the differential susceptibility hypotheses, it would have been desirable to expand measures of adversity in the direction of parental warmth and caregiver availability to better reflect the whole spectrum ranging from superior to poor environmental conditions [66]. Forth, since we assessed only adversity during childhood, we were unable to exclude confounding effects of recent negative life events, which may also have impact on present social behavior. Thus, future studies may investigate childhood, as well as recent adversity, in order to define the concrete contribution of these factors on social behavior and the development of psychopathology.

Together, the present study is the first to show that the association of empathic perspective-taking with childhood adversity is moderated by the 5-HTTLPR, and the transcriptional activity of the serotonin transporter gene. It therefore corroborates previous findings suggesting that the S-allele of the 5-HTTLPR conveys differential susceptibility to environmental cues, as does low transcriptional activity.

## Supporting information

**S1 Table. Data from genotyping, CTQ and IRI questionnaires for each participant.**
(PDF)

## Author Contributions

**Conceptualization:** Vera Flasbeck, Martin Brüne.

**Formal analysis:** Martin Brüne.

**Investigation:** Vera Flasbeck, Johanna Pakusch.

**Methodology:** Dirk Moser.

**Project administration:** Martin Brüne.

**Resources:** Robert Kumsta, Martin Brüne.

**Supervision:** Dirk Moser, Robert Kumsta.

**Writing – original draft:** Vera Flasbeck, Martin Brüne.

**Writing – review & editing:** Dirk Moser, Johanna Pakusch, Robert Kumsta, Martin Brüne.

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
