## [Decision Letter · Decision Letter 0]

9 Aug 2019

PONE-D-19-14573

The association between childhood maltreatment and empathic perspective taking is moderated by the 5-HTT linked polymorphic region. Another example of “differential susceptibility”

PLOS ONE

Dear Prof. Brüne,

Thank you for submitting your manuscript to PLOS ONE. After careful consideration, we feel that it has merit but does not fully meet PLOS ONE’s publication criteria as it currently stands. Therefore, we invite you to submit a revised version of the manuscript that addresses the points raised during the review process.

We would appreciate receiving your revised manuscript by Sep 23 2019 11:59PM. To enhance the reproducibility of your results, we recommend that if applicable you deposit your laboratory protocols in protocols.io, where a protocol can be assigned its own identifier (DOI) such that it can be cited independently in the future. For instructions see: http://journals.plos.org/plosone/s/submission-guidelines#loc-laboratory-protocols

We look forward to receiving your revised manuscript.

Kind regards,

Huiping Zhang

Academic Editor

PLOS ONE

Journal Requirements:

Reviewers' comments:

Reviewer's Responses to Questions

**Comments to the Author**

1. Is the manuscript technically sound, and do the data support the conclusions?

Reviewer #1: Yes

Reviewer #2: Partly

2. Has the statistical analysis been performed appropriately and rigorously? 

Reviewer #1: Yes

Reviewer #2: No

3. Have the authors made all data underlying the findings in their manuscript fully available?

Reviewer #1: Yes

Reviewer #2: Yes

4. Is the manuscript presented in an intelligible fashion and written in standard English?

Reviewer #1: Yes

Reviewer #2: Yes

5. Review Comments to the Author

Reviewer #1: This study focuses on the effect of 5-HTT linked polymorphic region in the association between childhood maltreatment and empathic perspective taking. In 279 female patients with BPD or healthy participants, childhood maltreatment and empathic abilities were measures using standardized questionnaires and genotypes in 5-HTT linked polymorphic region were determined and further categorized by different levles of transcriptional activities. Moderator analysis revealed that S-allele carriers and corresponding lower transcriptional activities showed significant response to childhood mental environmental impact in empathic perspective taking.

Previous studies mainly investigated the association negative environmental impact in empathic perspective taking while this study demonstrated the bidirectional empathic perspective taking outcome from childhood treatment.

Major:

1.How was the effect of recent major life events in adulthood evaluated for the 279 participants, which may serve as controls in this study.

2.Moderator analysis of CTQ total score, perspective taking and the BPD/ healthy participants.

Minor:

1.How does the CTQ questionnaire define age range of childhood?

2.Moderator analysis on three genotypes (Line 264-268) can be moved after moderator analysis on SS+SL vs LL (after line 244).

3.Typo: genotype should be genotypes in line 264.

4.In Table 3 and regarding description, the total n in the SS+SL group was stated as 176, however, only 175 samples can be identified.

5.In Table 3, LGSG group or SGSG group are also not presented in patients or healthy participants. Either remove the SASG group from the table or add these groups.

6.In the discussion (lines 287-291), the author’s statement regarding the benefit of emotional warmth and availability during childhood is confusing.

Reviewer #2: In this manuscript, authors investigate the role of 5-HTTLPR on the effects of child maltreatment on empathy. In a sample of 279 women (142 with BPD and 137 healthy controls), they studied the effects of 5-HTTLPR genotype on empathic perspective taking. The manuscript is well-written, but not rigorous in the study design and methods. Further, the conclusions made are premature, given the lack of replication and the numerous weaknesses identified.

1. There is a lack of power due to the small sample size of their cohort, particularly for the interaction analysis. Authors should include a power analysis to justify the sample size used in the study.

2. All subjects should have trauma information, including controls, since one of the main analysis performed is an interaction analysis.

3. There is no information about race. This should be also accounted for in the analysis.

4. Depression is highly prevalent among BPD. Analysis should account for depression as a covariate.

5. No information about healthy controls group is provided. What is the prevalence, if any, of disorders and medication listed Table 1 for BPD group.

6. No efforts for replication were made. Given the sample size, it is critical to conduct a replication analysis in an independent cohort.

7. In the largest meta-analysis studying the interaction between stress and 5-HTTLPR genotype in the context of depression, the Culverhouse et al found no association. Authors should discuss why studying 5-HTTLPR genotype is still relevant.

6. PLOS authors have the option to publish the peer review history of their article (what does this mean?). If published, this will include your full peer review and any attached files.

Reviewer #1: No

Reviewer #2: No

---

## [Author Response · Author response to Decision Letter 0]

6 Sep 2019

Revision 1

PONE-D-19-14573

The association between childhood maltreatment and empathic perspective taking is moderated by the 5-HTT linked polymorphic region. Another example of “differential susceptibility”

Journal Requirements:

Response: We revised the manuscript with regards to the journal requirements. We also add the study findings as supporting information.

Reviewer #1: 

This study focuses on the effect of 5-HTT linked polymorphic region in the association between childhood maltreatment and empathic perspective taking. In 279 female patients with BPD or healthy participants, childhood maltreatment and empathic abilities were measures using standardized questionnaires and genotypes in 5-HTT linked polymorphic region were determined and further categorized by different levles of transcriptional activities. Moderator analysis revealed that S-allele carriers and corresponding lower transcriptional activities showed significant response to childhood mental environmental impact in empathic perspective taking.

Previous studies mainly investigated the association negative environmental impact in empathic perspective taking while this study demonstrated the bidirectional empathic perspective taking outcome from childhood treatment.

Major:

1. How was the effect of recent major life events in adulthood evaluated for the 279 participants, which may serve as controls in this study.

Response: Unfortunately, we cannot exclude or control for effects of recent/present life events, because we did not assess traumatic experiences during adulthood. Since the reviewer raised an important point with this concern, we add this point to the limitation section. This part reads as follows: “Forth, since we assessed only adversity during childhood we were unable to exclude confounding effects of recent negative life events, which may also have impact on present social behavior. Thus, future studies may investigate childhood, as well as recent adversity, in order to define the concrete contribution of these factors on social behavior and the development of psychopathology.”

2. Moderator analysis of CTQ total score, perspective taking and the BPD/ healthy participants.

Response: We agree with the reviewer that is important to show that the associations found were not already coming up because of the BPD and HC groups. Therefore, we added another moderation analysis with Group (BPD/HC) as the moderator. We added the following part to the methods: “Another moderation analyses was performed for the SS, SL and LL Genotypes and for Group (BPD vs HC) as the moderator instead of Genotype. “

The following part was added to the results section: “We further aimed to examine whether the association of childhood trauma with perspective talking was related to diagnosis. Therefore, we performed the same moderation analysis but tested the factor group (BPD and HC) as the moderator. The model was also significant (F(5, 199) = 11.71, p < 0.001, R² = 0.2272), but the interaction of CTQ by group was not (Interaction b = 0.0437, t(199) = 0.81 , p = 0.419), suggesting that a diagnosis of BPD was not the sole factor impacting on the association of childhood trauma with empathic perspective taking “

We also extended the limitation section in the discussion: “However, since our main interest pertained to the influence of childhood adversity on trait empathy and its moderation by genotype, we were much less concerned with the presence or absence of specific effects of a diagnosis of BPD. In support of this idea, the moderation analysis with the moderator group (BPD/HC) did not show a specific effect of BPD on the association of childhood trauma with perspective taking.”

Minor:

1. How does the CTQ questionnaire define age range of childhood?

Response: The age range is not clearly defined by the questionnaire. The instruction on the questionnaire asks the participants to refer to their childhood and adolescence. This information was added to the methods: “Participants were asked to rate the occurrence of maltreatment on a 5-point Likert scale (1 = never, 5 = very often) pertaining to their childhood and youth.”

2. Moderator analysis on three genotypes (Line 264-268) can be moved after moderator analysis on SS+SL vs LL (after line 244).

Response: We agree with the reviewer and change the manuscript accordingly

3. Typo: genotype should be genotypes in line 264.

Response: Right, thanks for the hint!

4. In Table 3 and regarding description, the total n in the SS+SL group was stated as 176, however, only 175 samples can be identified.

Response: We verified the data shown in Table 3 several times, but could not find a mistake. 

5. In Table 3, LGSG group or SGSG group are also not presented in patients or healthy participants. Either remove the SASG group from the table or add these groups.

Response: We agree with the reviewer and decided to remove the SASG group.

6. In the discussion (lines 287-291), the author’s statement regarding the benefit of emotional warmth and availability during childhood is confusing.

Response: we revised the part in order to make it more clearly: “As noted by Belsky et al. [63], the mere absence of maltreatment is not equivalent to high-quality parenting. When applying this idea to the present findings, it can tentatively be hypothesized that emotional warmth and availability during early development may lead to even better-than-average perspective-taking abilities (in this case, the slopes of the regression lines shown in Fig 1 may diverge to a greater extent if extrapolated to the left).”

Reviewer #2:

In this manuscript, authors investigate the role of 5-HTTLPR on the effects of child maltreatment on empathy. In a sample of 279 women (142 with BPD and 137 healthy controls), they studied the effects of 5-HTTLPR genotype on empathic perspective taking. The manuscript is well-written, but not rigorous in the study design and methods. Further, the conclusions made are premature, given the lack of replication and the numerous weaknesses identified.

1. There is a lack of power due to the small sample size of their cohort, particularly for the interaction analysis. Authors should include a power analysis to justify the sample size used in the study.

Response: Following up on the reviewer’s concern, we added a power analysis for the interaction effect: “We conducted a power analysis for the interaction, i.e. the differences between slopes for the moderation model with the genotypes SS+SL and LL. Therefore, we used G*Power Version 3.1.9.2. (56). For the calculation of the achieved power with the present sample for the interaction effects we applied the option “t-test-linear bivariate Regression, two groups, difference between slopes”, which showed, that we reached a statistical power of α = 0.749 with a sample size of 205 participants.”

We also extended the discussion accordingly: “In any event, even though the statistical power for detecting significant interaction effects was sufficiently large with α = 0.75, it, it is warranted to replicate the study in an independent, and preferably larger, sample.”

In addition, we would like to mention that other recent work utilized comparable sample sizes (for example Nardi et al., 2013; Owens et al., 2012; Wagner et al., 2009, for review on 5-HTTLPR and psychopathology see Kenna et al., 2012)

2. All subjects should have trauma information, including controls, since one of the main analysis performed is an interaction analysis. 

Response: The moderation analyses were conducted for the whole sample, e.g. controls and BPD (except missing data). Participants of both groups received the CTQ (see also Table 2) 

3. There is no information about race. This should be also accounted for in the analysis.

Response: The reviewer is absolutely right with expressing concern regarding the ethnical background in a genetic study. We added the information to the material and methods – participant section. “Regarding the ethnical background, 95.2 % were Caucasians, 4.4 % originated from the Middle East (mainly of Turkish origin) and 0.4 % from the Far East (Vietnamese).”

4. Depression is highly prevalent among BPD. Analysis should account for depression as a covariate.

Response: We agree with the reviewer and added moderation analyses, which include the covariate BDI score.

The methods were extended as following: “In addition, we used the Beck’s Depression Inventory to assess the self-rated level of depressivity [54 […] “In order to examine the impact of depressivity, we calculated additional moderation analyses with the additional covariate “BDI score” for the moderators “SS+SL vs. LL”, “SS, SL and LL” and transcriptional activity.”

We extended the results accordingly (see also Table 2): “Patients differed in age and IQ from control participants and reported more severe experiences of childhood maltreatment as well as more depressive symptoms.” […] “Since depressivity is highly prevalent among individuals suffering from BPD, we repeated the moderation analyses and included also the BDI score as a covariate into the calculation. Here, the same results were obtained as in the previous analyses without the covariate BDI (moderator = SS+SL vs. LL: the model (F(6, 196) = 9.20, p < 0.001, R² = 0.2198) and the interaction (b = -0.0857, t(196) = -2.38 , p = 0.018) were significant; moderator = SS vs.SL vs. LL: the model was significant (F(6, 196) = 8.81, p < 0.001, R² = 0.2124), the interaction showed only a tendency (b = -0.0479, t(196) = -1.84 , p = 0.067); moderator TA group: the model (F(6, 196) = 9.43, p < 0.001, R² = 0.2239) and interaction (b = -0.0649, t(196) = -2.63 , p = 0.009) were significant).

We also discussed the association with depressivity more extensively, as raised by Reviewer (Comment 7) and referred to our results concerning depression: “[…] In our study, the inclusion of depression as a covariate did not affect the interaction, which suggests that the effect on perspective taking was not due to symptoms of depression. This further implies that our study does also not support the link between 5-HTTLPR, stress and depression.[…]”

5. No information about healthy controls group is provided. What is the prevalence, if any, of disorders and medication listed Table 1 for BPD group.

Response: We agree with the reviewer that no information was given. We added the following sentence to the methods: “The control participants were free of medication and psychiatric disorders.”

6. No efforts for replication were made. Given the sample size, it is critical to conduct a replication analysis in an independent cohort.

Response: We absolutely agree with the reviewer that a replication in a larger sample is required and included this point in the discussion.

Discussion: “In any way, since the power for detecting significant interaction was 0.749, it is warranted to replicate the study in an independent, and preferably larger, sample.

7. In the largest meta-analysis studying the interaction between stress and 5-HTTLPR genotype in the context of depression, the Culverhouse et al found no association. Authors should discuss why studying 5-HTTLPR genotype is still relevant.

Response: We are grateful for the advice to extend the discussion to the depression debate and added the following section to the discussion: “With regard to depression, the exact role of the 5-HTTLPR in the development of depression is unclear. This was shown by recent meta-analyses, which reported on the one side an association of stressful live events and depression in SS-homozygotes or S-carrying heterozygotes [42, 43], and on the other side other studies found no significant association [31, 44]. Culverhouse and colleagues found a significant main effect of sex and life stressor (high risk factor for depression), but they did not found an effect of genotype on the association of stress and depression, even if they included only studies with large sample sizes [44]. The authors concluded that if any interaction would exist, it would not be a generalizable effect, only detectable in limited situations and of modest effect size. In our study, the inclusion of depression as a covariate did not affect the interaction, which suggests that the effect on perspective taking was not due to symptoms of depression. This further implies that our study does also not support the link between 5-HTTLPR, stress and depression. Eventually, the 5-HTTLPR, together with stress, may induce stress and emotion processing impairments (as described above), which lead only in a subset of individuals to the development of depression. This subset may bear additional risks, which are currently not in the focus of interaction studies. One potential factor could be the transcriptional activity of the serotonin transporter gene, which could be altered by epigenetic processes or other SNPs, as for example the rs25531 [33, 34, 67].

---

## [Decision Letter · Decision Letter 1]

19 Nov 2019

PONE-D-19-14573R1

The association between childhood maltreatment and empathic perspective taking is moderated by the 5-HTT linked polymorphic region. Another example of “differential susceptibility”

PLOS ONE

Dear Prof. Brüne,

Thank you for submitting your manuscript to PLOS ONE. After careful consideration, we feel that it has merit but does not fully meet PLOS ONE’s publication criteria as it currently stands. Therefore, we invite you to submit a revised version of the manuscript that addresses the points raised during the review process.

There are some concerns about the organization of the content in the manuscript. For example, some data analyses and the obtained results were placed in the Result section without describing the method in the Methods section.

We would appreciate receiving your revised manuscript by Jan 03 2020 11:59PM. To enhance the reproducibility of your results, we recommend that if applicable you deposit your laboratory protocols in protocols.io, where a protocol can be assigned its own identifier (DOI) such that it can be cited independently in the future. For instructions see: http://journals.plos.org/plosone/s/submission-guidelines#loc-laboratory-protocols

We look forward to receiving your revised manuscript.

Kind regards,

Huiping Zhang

Academic Editor

PLOS ONE

Reviewers' comments:

Reviewer's Responses to Questions

**Comments to the Author**

1. If the authors have adequately addressed your comments raised in a previous round of review and you feel that this manuscript is now acceptable for publication, you may indicate that here to bypass the “Comments to the Author” section, enter your conflict of interest statement in the “Confidential to Editor” section, and submit your "Accept" recommendation.

Reviewer #1: (No Response)

Reviewer #3: All comments have been addressed

2. Is the manuscript technically sound, and do the data support the conclusions?

Reviewer #1: Partly

Reviewer #3: Partly

3. Has the statistical analysis been performed appropriately and rigorously? 

Reviewer #1: Yes

Reviewer #3: Yes

4. Have the authors made all data underlying the findings in their manuscript fully available?

Reviewer #1: Yes

Reviewer #3: Yes

5. Is the manuscript presented in an intelligible fashion and written in standard English?

Reviewer #1: Yes

Reviewer #3: No

6. Review Comments to the Author

Reviewer #1: Considering the study focuses on childhood adversity instead of lifetime adversity, better definition in age and evidence of adversity only in childhood are critical in the study design.

Reviewer #3: In this manuscript, Flasbeck et al. investigated whether, or not, the genetic variations in the 5HTTLP region (or SLC6A4) modify the effect of childhood trauma exposure on the empathy. To test this hypothesis, they collected the 205 sample of 142 BPD patients and 137 controls, and they measured PT and CTQ scores related with empathy and childhood trauma exposure, respectively. Also, they genotyped two genetic variations including 1) short/long alleles of 5HTTLPR and 2) rs25531 (A or G alleles). They conducted moderation analysis and they found that the effect of childhood trauma exposure on the empathy (PT score) among the carriers with specific alleles (short-5HTTLPR or A allele of rs25531) significantly differs from its effect among the non-carriers. Therefore, they concluded that the genetic variations of the 5HTTLP modifies the risk effect of childhood trauma exposure on the empathy.

Major comments:

• There were three moderation analyses with the same outcome and predictor variables by the different variables: 1) S allele carriers vs non-carriers; 2) transcriptional activity (L/A genotype vs others); and 3) the BPD status. Although all the summary statistics were written down in the result section, it would be much better for the audience if they summarize the statistics into a table.

• The author did power calculation using G*Power, but they did not explain about how they computed in detail at all. They should explicitly explain the parameter values (e.g. expected effect size and significance threshold) in the method section.

Minor comments:

• There are many different statistical tests in the manuscript. I think the main analysis is the moderation analysis. So, it would be nice if author better shows the three main moderation analysis models at once in the method section. It could be a formula format or a table. This will really improve understanding of this study.

• I think the full name of MANOCOVA should be defined in the page 8 (line number: 165).

• I think you meant the moderator (M: low/low, high/low and high/high), rather than (M: low/low, high/low and low/low), in the page 13 (line number: 253).

• I am not sure why the subtitle, “Genotyping” was written in Italic font in the page 7 (line number 145).

• Grammar error. “by calculation the correlation” in the page 8 (line number 176).

• What does “X2(2)” mean in the page 10 (line number: 204)?

• Author wrote why they divided the sample into S carriers vs. LL carriers in the result section in the page 10 (line number: 205-206). I think this should go into the method section. Also, it would be nice if author briefly explain about the rationale behind this stratification by S allele.

• I found that something should go into the discussion section other than in the result. The sentence in the page 12 (line number: 231-232) is the interpretation about the result from the moderation test.

• It was not described how the moderation analysis for the three genotypes was conducted. Did you use the additive model? It should be described in the method section.

• It would be nice if the authors show the LD information between the S/L variations and rs25531. Also, the allele frequencies of these variations in general population could be relevant information, which may exist in the gnomAD (https://gnomad.broadinstitute.org/) database.

7. PLOS authors have the option to publish the peer review history of their article (what does this mean?). If published, this will include your full peer review and any attached files.

Reviewer #1: No

Reviewer #3: No

---

## [Author Response · Author response to Decision Letter 1]

25 Nov 2019

Revision 2

PONE-D-19-14573

The association between childhood maltreatment and empathic perspective taking is moderated by the 5-HTT linked polymorphic region. Another example of “differential susceptibility”

Reviewer #1: 

Considering the study focuses on childhood adversity instead of lifetime adversity, better definition in age and evidence of adversity only in childhood are critical in the study design

Response: We are grateful for the reviewer’s advice. Accordingly, we explicitly make the distinction between childhood trauma and other adverse life events in the discussion. In addition, we now point out, throughout the manuscript, that the present study deals with early adversity. 

Reviewer #2:

Major comments: 

* There were three moderation analyses with the same outcome and predictor variables by the different variables: 1) S allele carriers vs non-carriers; 2) transcriptional activity (L/A genotype vs others); and 3) the BPD status. Although all the summary statistics were written down in the result section, it would be much better for the audience if they summarize the statistics into a table. 

Response: We are thankful for the comment and created table 5, which shows the moderator variables, covariates and statistics of the moderation analyses conducted (page 15). The table was a great advice, since results are more comprehensive and accessible for the reader. We therefore deleted the results of analyses with BDI as an additional covariate, since it can be found in the table. 

Table 5. Summary of moderation analyses performed for the predictor variable “CTQ” (total score) and the outcome variable “perspective taking”. The table shows the moderator variables and covariates used and the respective model statistics and interactions between the predictor and moderator variables. The last two columns show tests for differential susceptibility, i.e. differences between slopes and differences from zero.

Moderator Covariate Model Interaction moderator*CTQ Differences of slopes from 0 Differences between slopes

SS+SL vs. LL age, IQ F(5, 199) = 6.48, p < 0.001, R² = 0.1401 b = -0.0810, t(199) = -2.16 , p = 0.032 SS+SL b = -0.109, SE = 0.023, p < 0.001 t = 3.08, p = 0.002 

 LL: b = -0.028, 

SE = 0.031, p = 0.378

 age, IQ, BDI (F(6, 196) = 9.20, p < 0.001, R² = 0.2198 b = -0.0857, t(196) = -2.38 , p = 0.018 

SS vs. SL vs. LL age, IQ F(5, 199) = 6.77, p < 0.001, R² = 0.1374 b = -0.0497, t(199) = -1.84 , p = 0.067 

 age, IQ, BDI b = -0.0479, t(196) = -1.84 , p = 0.067 

TA groups: low/low, high/low, high/high age, IQ F(5, 199) = 6.77, p < 0.001, R² = 0.1453 b = -0.0624, t(199) = -2.44 , p = 0.015 high/high: b = -0.0378, SE = 0.0254, p = 0.1389 high/high vs. low/low: t = 3.30, p = 0.001

 high/low: b = -0.0808, SE = 0.0189, p < 0.001; high/low vs. low/low: t = 2.00, p = 0.047

 low/low: b = -0.1237, SE = 0.0262, p < 0.001 high/high vs. high/low group t = 2.01, p = 0.046

 age, IQ, BDI F(6, 196) = 9.43, p < 0.001, R² = 0.2239 b = -0.0649, t(196) = -2.63 , p = 0.009 

BPD vs. HC age, IQ F(5, 199) = 11.71, p < 0.001, R² = 0.2272 b = 0.0437, t(199) = 0.81 , p = 0.419 

* The author did power calculation using G*Power, but they did not explain about how they computed in detail at all. They should explicitly explain the parameter values (e.g. expected effect size and significance threshold) in the method section. 

Response: We extended the section on the power calculation as follows: Page 8 line 161-165: “We conducted a power analysis for interaction effects, i.e. the differences between slopes for the moderation model with the genotypes SS+SL and LL, using G*Power, Version 3.1.9.2. [59]. Power calculation for the current sample of 205 participants was determined by the following model: t-test-linear bivariate regression, two groups, difference between slopes, with α set at 0.05. Standard deviation of the residuals, the sample size and the difference between the slopes were also considered. Accordingly, the statistical power coefficient was 0.75. “ 

Minor comments: 

* There are many different statistical tests in the manuscript. I think the main analysis is the moderation analysis. So, it would be nice if author better shows the three main moderation analysis models at once in the method section. It could be a formula format or a table. This will really improve understanding of this study. 

Response: The authors completely agree with the reviewer that the statistical methods section was very long and difficult to follow. We therefore added another table showing the moderation analyses (page 10)

Table 2. Summary of moderation analyses conducted. The predictor and outcome variables remained constant across calculations whereas the moderator and control variables were exchanged. 

Predictor Outcome Moderator Covariates

CTQ 

total score

 Perspective 

taking

 SS+SL vs. LL age, IQ

 age, IQ, BDI

 SS vs. SL vs. LL age, IQ

 age, IQ, BDI

 TA groups: low/low, high/low, high/high age, IQ

 age, IQ, BDI

 BPD vs. HC age, IQ

* I think the full name of MANOCOVA should be defined in the page 8 (line number: 165). 

Response: We changed this sentence accordingly (now page 9 line 179): “In order to investigate the effect of the genotype, we calculated a multivariate analysis of covariance (MANCOVA) with the covariates IQ and age and the between-subject factor genotype (SS+SL vs. LL) and the independent variables were the IRI scores.”

* I think you meant the moderator (M: low/low, high/low and high/high), rather than (M: low/low, high/low and low/low), in the page 13 (line number: 253). 

Response: Correct, this was an error, thank you for mentioning!

* I am not sure why the subtitle, "Genotyping" was written in Italic font in the page 7 (line number 145). 

Response: Sorry, this shouldn’t be italic and was corrected.

* Grammar error. "by calculation the correlation" in the page 8 (line number 176). 

Response: Thank you for the hint, we improved the sentence to “by calculating the correlation of…”

* What does "X2(2)" mean in the page 10 (line number: 204)? 

Response: The (2) stands for the df of the Chi square test, we changed it to: “(X² = 2.84; p = 0.241; df = 2; Table 4)”

* Author wrote why they divided the sample into S carriers vs. LL carriers in the result section in the page 10 (line number: 205-206). I think this should go into the method section. Also, it would be nice if author briefly explain about the rationale behind this stratification by S allele. 

Response: We transferred the section into the methods section and added the following part explaining the rationale (page 8 line 166-172): “In accordance with previous studies, we divided the sample into S-carriers (SS+SL pooled) and LL-carriers (e.g. [60-62]). This approach was justified, because previous studies reported no differences between SS and SL-carriers with regard to personality traits, suggesting a dominant-recessive type of association of the S-allele with personality (21). 

Similarly, following previous research (e.g., [37]), subjects were further divided into groups according to the ”transcriptional activity“ (TA) of the rs25531, which is reported in detail in Table 4. “ 

The results were changed to (page 11 line 223-226): “The division into S or LL-carriers resulted in n = 103 LL-carriers and n = 176 S-carriers. Another group formation was based on the rs25531, which offers the opportunity to build “transcriptional activity“ (TA) groups, (see Table 4).”

* I found that something should go into the discussion section other than in the result. The sentence in the page 12 (line number: 231-232) is the interpretation about the result from the moderation test. 

Response: The authors agree with the reviewer and deleted the interpretation in the results.

* It was not described how the moderation analysis for the three genotypes was conducted. Did you use the additive model? It should be described in the method section. 

Response: The moderation analysis of the three genotypes can be found on page 9 line 194-196 and now in the table 2.

* It would be nice if the authors show the LD information between the S/L variations and rs25531. Also, the allele frequencies of these variations in general population could be relevant information, which may exist in the gnomAD (https://gnomad.broadinstitute.org/) database. 

Response: We inserted the information into the introduction (page 3 line 69-71, page 4 line 73-76): “However, this polymorphic variation does not generally occur more frequently in clinical samples compared to the general population [30-32]. The general population is heterozygous, whereas the LL-genotype is less common, and the SS-variant relatively rare, in part depending on ethnicity [33, 34]. In addition, controversy exists about the effect of rs25531, a SNP within the 5-HTTLPR repetitive element, with the A-variant of the L-allele being associated with greater transcriptional activity and thus more efficient serotonin turnover [35-37], whereby a linkage disequilibrium between 5-HTTLPR and rs25531 has been described, with the rarer G-variant of rs25531 occurring more frequently together with the L-allele than with the S-allele [36, 38].

---

## [Editor Report · Decision Letter 2]

6 Dec 2019

The association between childhood maltreatment and empathic perspective taking is moderated by the 5-HTT linked polymorphic region. Another example of “differential susceptibility”

PONE-D-19-14573R2

Dear Dr. Brüne,

We are pleased to inform you that your manuscript has been judged scientifically suitable for publication and will be formally accepted for publication once it complies with all outstanding technical requirements.

With kind regards,

Huiping Zhang

Academic Editor

PLOS ONE
---

## [Editor Report · Acceptance letter]

10 Dec 2019

PONE-D-19-14573R2 

The association between childhood maltreatment and empathic perspective taking is moderated by the 5-HTT linked polymorphic region. Another example of “differential susceptibility” 

Dear Dr. Brüne:

I am pleased to inform you that your manuscript has been deemed suitable for publication in PLOS ONE. Congratulations! Your manuscript is now with our production department. 

With kind regards,

on behalf of

Dr. Huiping Zhang 

Academic Editor

PLOS ONE